# A novel technique for olfactory bulb measurements

**Akshita Joshi**[1]\*, **Divesh Thaploo**[1], **Xiaoguang Yan**[1], **Theresa Herrmann**[1], **Hudaa Alrahman Khabour**[2], **Thomas Hummel**[1]

**1** Smell and Taste Clinic, Department of Otorhinolaryngology, TU Dresden, Dresden, Germany, **2** Jordan University of Science and Technology, Irbid, Jordan

\* joshiakshita93@gmail.com

## Abstract

### Background

To introduce new ways to calculate OB volumes, checking their validity and comparing them to already established technique i.e. OB volumetric based on manual segmentation of OB boundaries.

### Methods

Two approaches were used to calculate OB volumes (1) Manual Segmentation using planimetric manual contouring; (2) Box-frame method, calculating the parameters based on a box placed around the OB.

### Results

We calculated OB volumes using both techniques and found comparable outcomes. High inter-observer reliability was found for volumes calculated by both observers. For manual segmentation, Cronbach's alpha (α) was 0.91 and 0.93 for right and left OB volume, respectively, whereas for the box-frame method α was 0.94 and 0.90 for right and left OB, respectively.

### Conclusions

The simple box-frame method of OB volume calculation appears reliable. Its results are comparable to an established technique.

## Introduction

The olfactory bulb (OB) is a highly significant structure in the processing of olfactory information. It is the first relay station from the peripheral olfactory system to higher order processing of olfactory information. In animals, OB continuously replace its local GABAergic interneurons which signifies [1–3] continuous generation of new neurons throughout lifetime [4].

**Data Availability Statement:** All relevant data are within the manuscript and its Supporting Information files.

**Funding:** The authors received no specific funding for this work.

**Competing interests:** The authors have declared that no competing interests exist.

From the sub-ventricular zone (SVZ), the OB receives progenitor cells through the rostral migration stream, which have the property of differentiation [4]. These newly born adult cells further integrate into an already existing OB neural network, hence adapting its function [5].

Less is known about the plastic nature of the OB in humans. Its regenerative property in humans is still a topic of debate. A study by Bergmann et al., focusing on the age of OB neurons in humans concluded that age of the OB neurons equals the age of an individual and that less than 1% of OB neurons are replaced in one's entire lifetime [6]. However, other groups reported indications for major regenerative activity in the OB [7]. In addition, the influx of neurons from the SVZ to the OB had been described in humans [8] which compares to animals [2].

Humans have varied OB volumes, which had been hypothesized to depend on synaptic input from olfactory receptor neurons [9, 10] In healthy subjects, OB volume was found to positively correlate with measured olfactory function, and varying with age [11–13]. OB volume varies in subjects with different olfactory pathologies. For example, subjects with congenital anosmia may have under-developed or no OB, whereas reduced OB volume was reported in subjects with post-infectious and post-traumatic olfactory loss [14]. As an exception to this rule, Weiss et al. reported normal olfactory functioning in women who did not have clear and distinct OB [14–16].

The OB volume is of clinical importance to gauge olfactory function [13, 17, 18]. As reported, change in OB volume correlates well with odor threshold and odor identification [19]. Moreover, because assessment of OB volume requires manual delineation, it is time-consuming and needs specific training of observers. Hence, OB volume measurements are typically not used in routine examinations of patients with olfactory loss. This might change with the availability of tools allowing reliable but less investigator-biased and faster OB volume measurement. Hence, the aim of the present study was to introduce a new way to calculate OB volumes, examining (1) its test- retest reliability and (2) validity, comparing them to the established technique, i.e. OB volumetric based on manual segmentation of OB boundaries (3) checking usability of the new technique by experts and non- experts.

## Methods

### Subjects

To calculate OB volumes, 52 subjects underwent magnetic resonance imaging (MRI) of the brain. All participating subjects visited the Smell and Taste Clinic at the Department of Otorhinolaryngology, University Hospital Carl Gustav Carus (Dresden, Germany) and were clinically diagnosed with smell loss. The local Ethics Committee approved the study. All subjects provided written informed consent and were tested for their orthonasal olfactory functioning using the "Sniffin' Sticks" test battery [20] which comprises three olfactory tests: olfactory threshold for phenyl ethyl alcohol (a rose-like odor), odor discrimination and odor identification. These tests were used to categorize olfactory loss patients as being either functionally anosmic, hyposmic or normosmic [21].

### MRI acquisition

MRI data were acquired on a 3 Tesla scanner (model Prisma; Siemens, Erlangen, Germany). For the T2 weighted sequence a 32-channel head coil was used. The scanning parameters were: repetition time (TR) = 1500 ms; echo time (TE) = 78 ms; flip angle = 150˚; slice thickness = 1mm; field of view matrix = 256 x 320.

**Measurement of OB volume.** OB volumes (shown in Fig 1B) were calculated using two methods.

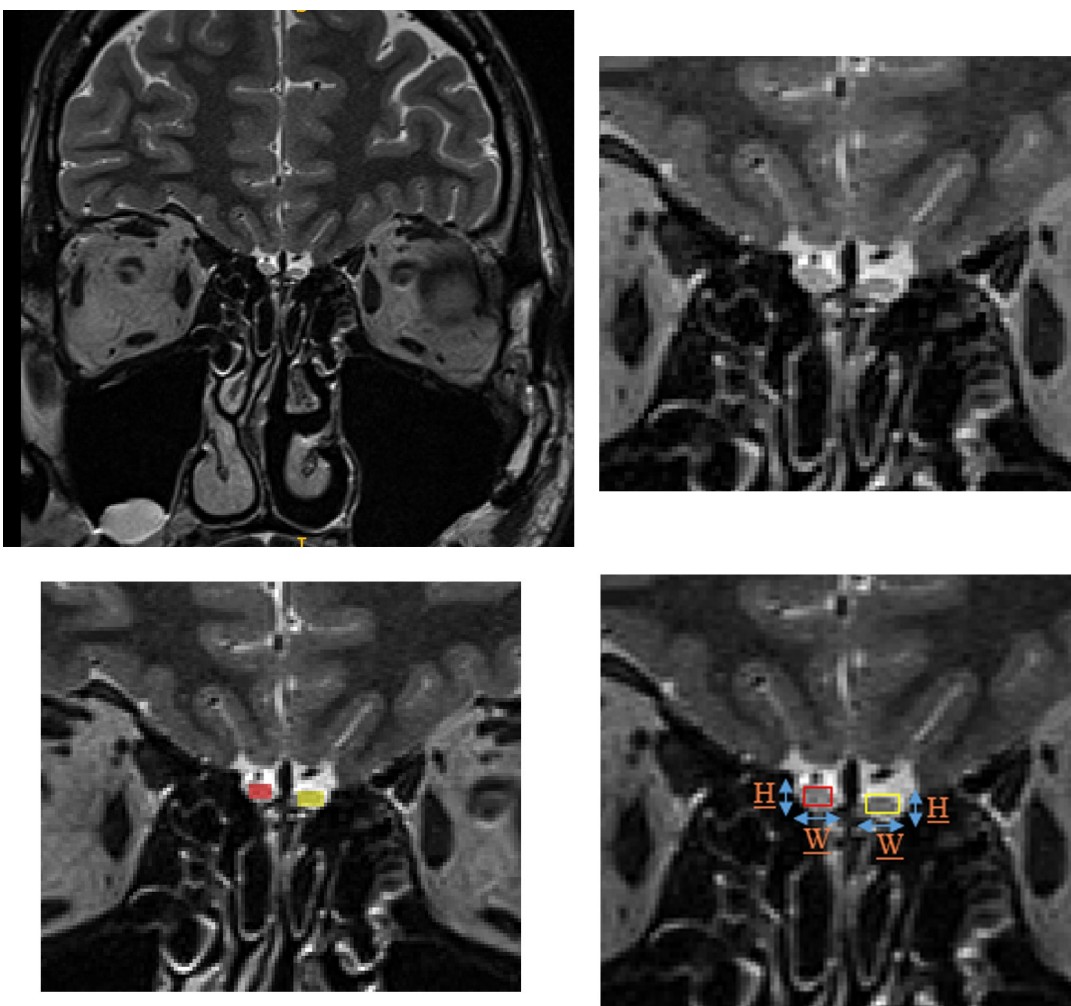

**Fig 1.** (A) Whole brain MR scan from a single subject. (B) right and left OB of the subject. (C) MS approach by plane manual contouring technique. (D) BF approach, with H = height and W = width depiction.

**Manual segmentation method (MS).** AMIRA 3D visualization and modeling system (Visage Imaging, Carlsbad, USA) was used to calculate the volume of right and left OB using the planimetric manual contouring (PMC) technique (surface in mm$^2$) (Fig 1A and 1C). The OB sequence included acquisition of 1 mm thick T2- weighted fast spin images, in the coronal plane that covers middle and anterior portions of the skull base. A standardized PMC protocol was applied to all scans [22]. Firstly, number of slices with clear visibility of the OB were selected. On each successive slice of brain, contours on left and right side of OB were manually drawn. The proximal end of the OB was defined by the abrupt change in the diameter at the beginning of the olfactory tract [22, 23].Two trained observers blind to the diagnosis and clinical characteristics of the subjects, calculated the volumes (in mm$^3$).

**Box- frame method (BF).** ITK-SNAP (version 3.8.0, University of Pennsylvania & University of Utah, www.itksnap.org) [24] was used for the alternative calculations of OB volumes. Firstly, the number of slices with distinct visibility of the OB was noted down. Further, the slice having the most visible voxels for both right and left side was chosen as the standard slice (in most cases it was the central slice). As the OB shape varies between individuals, we framed

a box on it as shown in Fig 1A and 1D. Annotations were drawn on the standard slice using Image annotation tool. With the help of this tool, we calculated the width (w) and height (h) by physically drawing a line between two extreme points of OB. For calculation of box volume, the length (l) was calculated by selecting the total number of slices which showed clear and distinct OB, multiplied by the slice thickness (1mm) ($V = l^*w^*h$, in $mm^3$). Two expert observers (AJ, XY), blind to the subject's condition calculated the volumes of right and left OBs. When the difference exceeded 10%, a third expert observer calculated the volumes again. After input of the third observer, two closest volumes with less than 10% difference were selected.

The idea for proposing the BF approach was also its usability by non-experts in neuroimaging. Accordingly, we checked its performance by non- expert observers who belonged to a different background with no imaging experience. They were well explained how the technique works and

**Table 1. Subject characteristics shown as mean ± standard deviation [SD] or number of subjects [N (%)].**

| | |
|---|---|
| Age (in years) | 56 ± 14 |
| Male/ female ratio | 15/ 32 |
| **Causes of olfactory loss** | |
| patients with idiopathic olfactory loss | N = 8 {17%} |
| patients with congenital olfactory dysfunction | N = 3 {6%} |
| patients with post- viral olfactory loss | N = 36 {77%} |
| **OB results using the Manual Segmentation**: | |
| Volume of right OB (Observer 1) (in $mm^3$) | 21.52 ± 11.42 |
| Volume of right OB (Observer 2) (in $mm^3$) | 19.25 ± 10.67 |
| Volume of left OB (Observer 1) (in $mm^3$) | 22.73 ± 13.11 |
| Volume of left OB (Observer 2) (in $mm^3$) | 20.44 ± 12.11 |
| **OB results using the Box- Frame method (expert)** | |
| Volume of right OB (Observer 1) (in $mm^3$) | 34.34 ± 18.46 |
| Volume of right OB (Observer 2) (in $mm^3$) | 32.96 ± 17.51 |
| Volume of left OB (Observer 1) (in $mm^3$) | 32.38 ± 17.53 |
| Volume of left OB (Observer 2) (in $mm^3$) | 31.52 ± 17.41 |
| **OB results using the Box- Frame method (non-expert)** | |
| Volume of right OB (Observer 1) (in $mm^3$) | 33.65 ± 17.78 |
| Volume of right OB (Observer 2) (in $mm^3$) | 39.24 ± 22.10 |
| Volume of left OB (Observer 1) (in $mm^3$) | 42.12 ± 24.46 |
| Volume of left OB (Observer 2) (in $mm^3$) | 42.71 ± 27.23 |
| **Olfactory test scores** | |
| TDI score | 17.91 ± 7.86 |
| Threshold score | 2.57 ± 2.54 |
| Discrimination score | 7.87 ± 3.40 |
| Identification score | 7.64 ± 3.49 |
| **Duration of smell loss** | |
| 0–2 years | 33 |
| 2–5 years | 8 |
| 5–10 years | 2 |
| >10 years | 4 |
| **Categorisation of participants** | |
| Functional anosmia | 23 |
| Hyposmia | 21 |
| Normosmia | 3 |

were asked to do the measurements in all of the subject population. Following the same rules, when the difference exceeded 10%, a third non-expert observer calculated the volumes again.

Out of the total 52 subjects, five subjects were excluded due to unclear OBs and lack of subject's information and finally, volumes of 47 subjects were analyzed and compared for left and right OB volumes. Out of them, 36 subjects had reduced olfactory functioning due to an infection in the upper respiratory tract (URTI), eight were diagnosed with idiopathic olfactory loss (ID) and three had congenital anosmia.

### Statistics

The Statistical Package for Social Sciences version 25.0 (IBM SPSS 25.0, Chicago, IL, USA) was used for statistical analysis. Table 1 shows the characteristic information for all subjects (means ± SD). A paired t-test was done to compare volumes of right and left OB as calculated by observers 1 and 2 using both methods. Furthermore, using Pearson correlation, inter-observer reliability was investigated for the volumes calculated by MS (AMIRA) and BF (ITK-SNAP) method. The level of significance was set at 0.05.

## Results

Mean volumes for right and left OB as measured by 2 observers using MS and BF-methods varied significantly ($p < 0.05$) with MS producing smaller volumes (Fig 2). Number of slices chosen by the 2 observers did not vary significantly for both methods. The mean number of slices for MS and BF methods were 6.3 and 6.8 respectively.

Positive correlation was found between OB volumes calculated by observer 1 and 2 for both methods: For MS, r = 0.84, p <0.01 (right OB) and r = 0.86, p <0.01 (left OB). For BF, r = 0.95, p<0.01 (right OB) and r = 0.89, p< 0.01 (left OB) (Table 2 and Fig 3).

Also, positive correlations were found between MS and BF methods (taking the average volumes measured by observer 1 and 2). For right OB, r = 0.73, p<0.01 and for left OB, r = 0.70, p<0.01 (Table 2).

High inter-observer reliability was found for volumes calculated by observers 1 and 2. For MS method, Cronbach's alpha (α) was 0.91 and 0.93 for right and left OB volume, respectively, whereas for the BF method α was 0.98 and 0.95 for right and left OB, respectively.

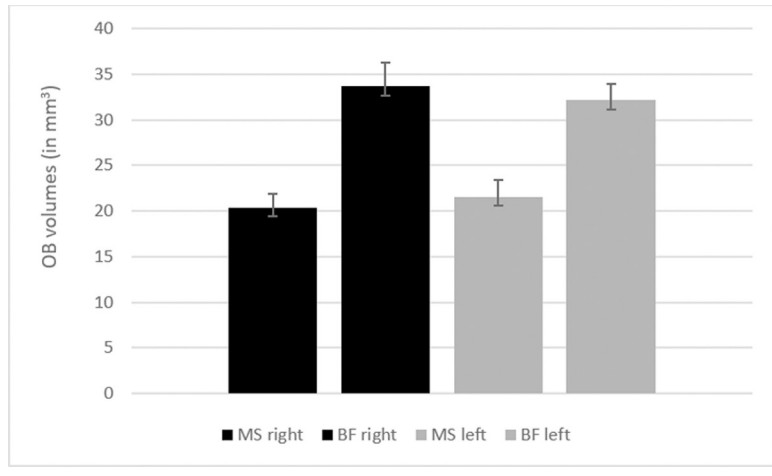

**Fig 2. Averaging from measurements done by both the expert observers, OB volumes in Mean ± SEM, measured by both methods: MS right = 20.38 ± 1.54; BF right = 33.65 ± 2.59; MS left = 21.58 ± 1.8; BF left = 31.95 ± 2.5.**

**Table 2. Correlations between right and left OB volumes obtained by expert observers (O1 = observer 1 and O2 = observer 2) and with different techniques (manual segmentation and box frame method).**

| | | Coefficients of correlation (r) | level of significance (p) |
|---|---|---|---|
| **O1 vs. O2:** Manual Segmentation | Right OB | 0.84 | <0.01 |
| **O1 vs. O2:** Box-frame | Right OB | 0.95 | <0.01 |
| **O1 vs. O2:** Manual Segmentation | Left OB | 0.86 | <0.01 |
| **O1 vs. O2:** Box-frame | Left OB | 0.90 | <0.01 |
| Manual Segmentation vs Box- frame | Right OB | 0.73 | <0.01 |
| Manual Segmentation vs Box- frame | Left OB | 0.70 | <0.01 |

For BF approach, inter- observer reliability was checked for measurements done by experts and non- experts. The Cronbach's alpha (α) for right OB was 0.82 and 0.83 for left OB. The results advantages its usability by non- experts or less trained as well.

## Discussion

In this study, we aimed to find an efficient, reliable yet less time-consuming method to calculate the OB volume. In fact, measurement time for the MS method was approximately 7–10 minutes whereas it takes only one minute for the BF method. Our study indicated that the BF approach provides reliable results which are in accordance with the results obtained from MS and when used by experts and non- experts.

So far, the MS of coronal slices is the most widely used method for volumetric measurements of the OB [25] Accuracy and reliability of MS method has been demonstrated clearly in previous studies [18, 26]. In the present study, we also followed up accuracy and reliability for the measurements made by the BF approach using ITK-SNAP software. This software was chosen for its user-friendly interface and free availability. However, many other software solutions could be used for this straight-forward technique. For the BF approach, intraclass coefficients of correlation between measurements of the two observers were at r = 0.96 for right OB and r = 0.89 for left OB. The results drawn from this new approach were comparable with the results obtained from MS approach with r = 0.84 for right OB and r = 0.86 for left OB.

The focus throughout the project was on the introduction of a method that can be clinically acceptable, with time demands being a major issue. This is important as OB volume is considered as a measure to evaluate the status of olfactory functioning. There has been evidence in support of how OB volume clinically describes the severity of olfactory loss. For example, in comparison to hyposmic patients, OB volumes were found to be smaller for anosmic subjects in olfactory loss, following infections of the upper respiratory tract or head trauma [27]. Importantly, OB volume also seems to be a predictor of recovery in patients with post-infectious olfactory loss [22]. Hence, the routine assessment of OB volume appears to be useful in patients with olfactory loss. This is more likely to be diagnostically implemented with the availability of a fast and convenient approach.

The present investigation also revealed that the internal consistency of measurements made with either method was excellent. Hence, it can be noted that the new BF method can be used as a clinically acceptable, efficient, reliable, easy and quick approach to calculate OB volumes. However, it has to be kept in mind that both MS and BF method remain subjective and voxel selection may vary depending on skills of the individual observers which requires some degree of training.

To conclude, the present results suggest that the BF method for OB volumetric is reliable and produces valid results, comparable to the results from MS. The new technique is a simple,

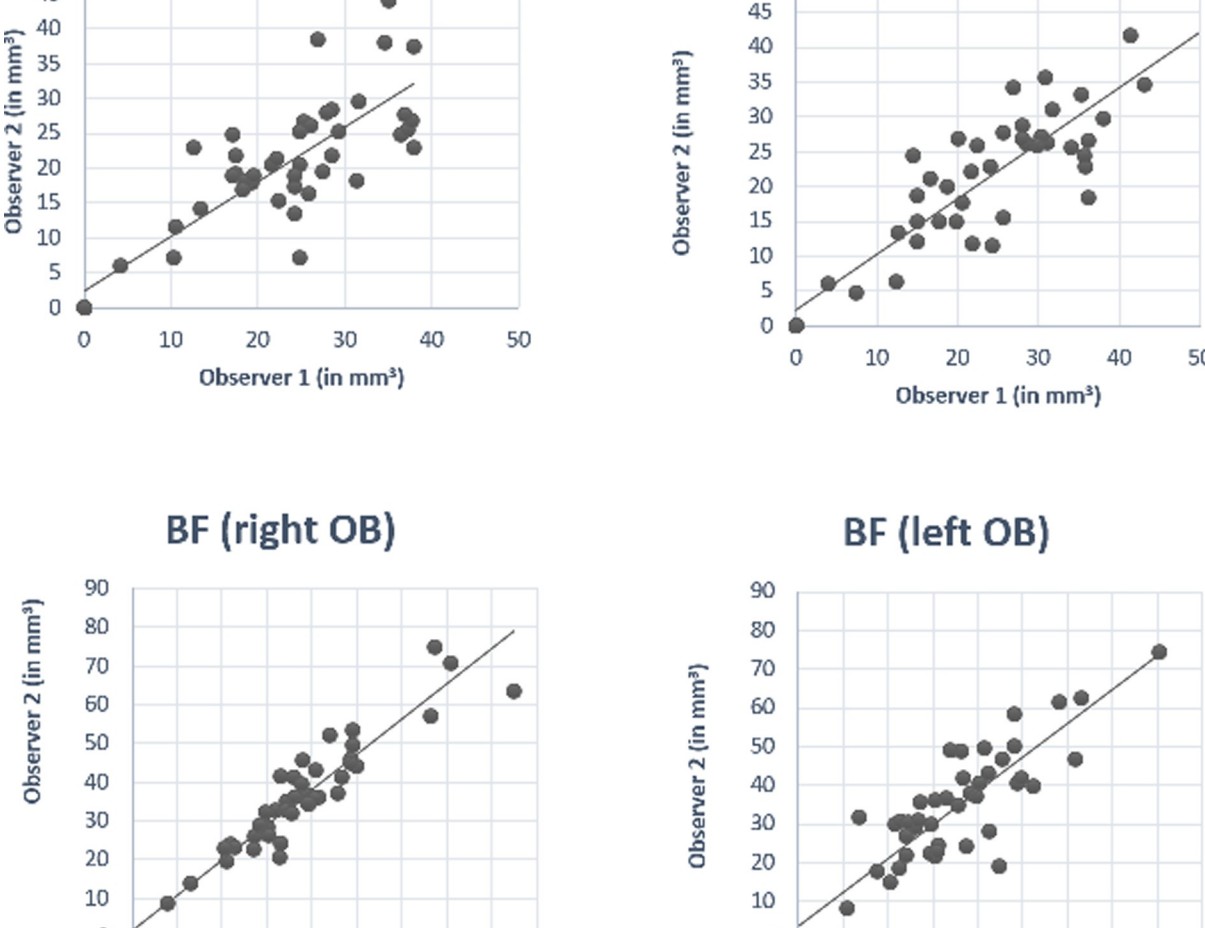

**Fig 3. Positive correlation between volumes (in mm³) measured by observer 1 and 2 for manual segmentation (MS) and box- frame methods (BF).** For MS, (A) r = 0.84, p <0.01 (right OB volume) and (B) r = 0.86, p <0.01 (left OB volume). For BF, (C) r = 0.95, p<0.01 (right OB volume) and (D) r = 0.90, p< 0.01 (left OB volume).

quick approach and may require less training than MS of the OB. It is hoped that this technique paves the road for the routine clinical assessment of OB volume in patients with olfactory loss.

## Supporting information

**S1 Data.**
(XLSX)

## Acknowledgments

We would like to thank Dmitriy Desser for his help in recommending the software. Many thanks to three non-expert observers for their time, efforts and cooperation.

## Author Contributions

**Conceptualization:** Akshita Joshi, Thomas Hummel.

**Data curation:** Akshita Joshi, Divesh Thaploo, Thomas Hummel.

**Formal analysis:** Akshita Joshi, Thomas Hummel.

**Funding acquisition:** Thomas Hummel.

**Investigation:** Thomas Hummel.

**Methodology:** Akshita Joshi, Divesh Thaploo, Xiaoguang Yan, Theresa Herrmann, Hudaa Alrahman Khabour, Thomas Hummel.

**Project administration:** Akshita Joshi, Thomas Hummel.

**Resources:** Thomas Hummel.

**Software:** Akshita Joshi, Divesh Thaploo, Xiaoguang Yan, Theresa Herrmann, Hudaa Alrahman Khabour, Thomas Hummel.

**Supervision:** Thomas Hummel.

**Validation:** Akshita Joshi, Divesh Thaploo, Xiaoguang Yan, Theresa Herrmann, Thomas Hummel.

**Visualization:** Xiaoguang Yan, Hudaa Alrahman Khabour, Thomas Hummel.

**Writing – original draft:** Akshita Joshi, Divesh Thaploo, Thomas Hummel.

**Writing – review & editing:** Akshita Joshi, Divesh Thaploo, Xiaoguang Yan, Thomas Hummel.

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
