## [Decision Letter · Decision Letter 0]

12 Oct 2020

PONE-D-20-23928

A novel technique for olfactory bulb measurements.

PLOS ONE

Dear Dr. Joshi,

Thank you for submitting your manuscript to PLOS ONE. After careful consideration, we feel that it has merit but does not fully meet PLOS ONE’s publication criteria as it currently stands. Therefore, we invite you to submit a revised version of the manuscript that addresses the points raised during the review process. As suggested by the reviewer, we strongly encourage you to benchmark the performance of your new technique with a couple of less-expert users who are unrelated to the manuscript. 

We look forward to receiving your revised manuscript.

Kind regards,

Matthieu Louis

Academic Editor

PLOS ONE

Journal Requirements:

"Akshita Joshi and Divesh Thaploo are supported by DAAD (Deutscher Akademischer

Austauschdienst / German Academic Exchange Service). The funding source had no role in

the study design, collection, analysis and interpretation of the data. Nor in the writing of the

manuscript, and in the decision to submit the paper for publication."

Reviewers' comments:

Reviewer's Responses to Questions

**Comments to the Author**

1. Is the manuscript technically sound, and do the data support the conclusions?

Reviewer #1: Yes

2. Has the statistical analysis been performed appropriately and rigorously? 

Reviewer #1: Yes

3. Have the authors made all data underlying the findings in their manuscript fully available?

Reviewer #1: Yes

4. Is the manuscript presented in an intelligible fashion and written in standard English?

Reviewer #1: Yes

5. Review Comments to the Author

Reviewer #1: In this manuscript, Joshi et al demonstrate a technique for delineating the olfactory bulb volume in MRI scans of the human brain. Whereas the standard technique takes a well-trained observer 10 minutes of tracing, the new technique requires less expertise and time. The demonstration that the two techniques give equivalent results is promising. However, given that the techniques were only applied by the study's co-authors, I wonder if this equivalence can be put to a more stringent test. Would the two techniques give equivalent results if applied by less expert observers? Since one of the claimed advantages of the box-frame technique is that it requires less expertise, it would make sense to determine whether box-frame gives equivalent performance for less-expert observers. I suggest that the authors ask other observers, preferably not co-authors of the study, to apply the two techniques. The value of the box-frame technique will be more compelling if its advantages generalize to non-authors.

6. PLOS authors have the option to publish the peer review history of their article (what does this mean?). If published, this will include your full peer review and any attached files.

Reviewer #1: **Yes: **Matt Smear

---

## [Author Response · Author response to Decision Letter 0]

20 Nov 2020

Dear Reviewers and Editors,

• Thank you very much for your time and comment on the manuscript. 

Reviewer #1: In this manuscript, Joshi et al demonstrate a technique for delineating the olfactory bulb volume in MRI scans of the human brain. Whereas the standard technique takes a well-trained observer 10 minutes of tracing, the new technique requires less expertise and time. The demonstration that the two techniques give equivalent results is promising. However, given that the techniques were only applied by the study's co-authors, I wonder if this equivalence can be put to a more stringent test. Would the two techniques give equivalent results if applied by less expert observers? Since one of the claimed advantages of the box-frame technique is that it requires less expertise, it would make sense to determine whether box-frame gives equivalent performance for less-expert observers. I suggest that the authors ask other observers, preferably not co-authors of the study, to apply the two techniques. The value of the box-frame technique will be more compelling if its advantages generalize to non-authors.

In response to the valuable comments raised by the reviewers; two non- experts belonging to completely different background with no imaging experience measured the bulb volumes using box- frame approach in total study population. Further, reliability was tested between measurements done by experts and non- experts. Cronbach alpha values shows that the technique is worth relying on and therefore supports its universal usability.

---

## [Editor Report · Decision Letter 1]

1 Dec 2020

A novel technique for olfactory bulb measurements.

PONE-D-20-23928R1

Dear Dr. Joshi,

We’re pleased to inform you that your manuscript has been judged scientifically suitable for publication and will be formally accepted for publication once it meets all outstanding technical requirements.

Kind regards,

Matthieu Louis

Academic Editor

PLOS ONE
---

## [Editor Report · Acceptance letter]

4 Dec 2020

PONE-D-20-23928R1 

A novel technique for olfactory bulb measurements. 

Dear Dr. Joshi:

I'm pleased to inform you that your manuscript has been deemed suitable for publication in PLOS ONE. Congratulations! Your manuscript is now with our production department. 

Kind regards, 

on behalf of

Dr Matthieu Louis 

Academic Editor

PLOS ONE